# Reactive Oxygen Species Produced by 5-Aminolevulinic Acid Photodynamic Therapy in the Treatment of Cancer

**DOI:** 10.3390/ijms24108964

**Published:** 2023-05-18

**Authors:** Pamela Pignatelli, Samia Umme, Domenica Lucia D’Antonio, Adriano Piattelli, Maria Cristina Curia

**Affiliations:** 1COMDINAV DUE, Nave Cavour, Italian Navy, Stazione Navale Mar Grande, Viale Ionio, 74122 Taranto, Italy; pamelapignatelli89p@gmail.com; 2Department of Medical, Oral and Biotechnological Sciences, “G. d’Annunzio” University of Chieti-Pescara, Via dei Vestini, 66100 Chieti, Italy; ummesamia95@gmail.com (S.U.); domenica.dantonio@unich.it (D.L.D.); 3Fondazione Villaserena per la Ricerca, Città Sant’Angelo, 65013 Pescara, Italy; 4Casa di Cura Villa Serena, Città Sant’Angelo, 65013 Pescara, Italy; 5School of Dentistry, Saint Camillus International University for Health Sciences, 00131 Rome, Italy; apiattelli51@gmail.com; 6Facultad de Medicina, UCAM Universidad Católica San Antonio de Murcia, Guadalupe, 30107 Murcia, Spain

**Keywords:** ROS, 5-aminolevulinic acid, PDT, cancer, cell death

## Abstract

Cancer is the leading cause of death worldwide and several anticancer therapies take advantage of the ability of reactive oxygen species to kill cancer cells. Added to this is the ancient hypothesis that light alone can be used to kill cancer cells. 5-aminolevulinic acid-photodynamic therapy (5-ALA-PDT) is a therapeutic option for a variety of cutaneous and internal malignancies. PDT uses a photosensitizer that, activated by light in the presence of molecule oxygen, forms ROS, which are responsible for the apoptotic activity of the malignant tissues. 5-ALA is usually used as an endogenous pro-photosensitizer because it is converted to Protoporphyrin IX (PpIX), which enters into the process of heme synthesis and contextually becomes a photosensitizer, radiating a red fluorescent light. In cancer cells, the lack of the ferrochelatase enzyme leads to an accumulation of PpIX and consequently to an increased production of ROS. PDT has the benefit of being administered before or after chemotherapy, radiation, or surgery, without impairing the efficacy of these treatment techniques. Furthermore, sensitivity to PDT is unaffected by the negative effects of chemotherapy or radiation. This review focuses on the studies done so far on 5-ALA-PDT and its efficacy in the treatment of various cancer pathologies.

## 1. Introduction: 5-Aminolevulinic Acid Photodynamic Therapy

Cancer is the leading cause of death worldwide [1]. Estimates of Incidence and Mortality Worldwide for 36 Cancers in 185 Countries. It is conventionally treated by surgery, radiation therapy, and chemotherapy, but such treatments may be fast-acting and cause adverse effects on normal tissues. Alternative and innovative methods of cancer treatment with the least side effects and improved efficiency are being encouraged [2]. Radiotherapy is the single most effective and inexpensive non-surgical treatment for cancer, but it presents numerous side effects [3]. This includes acute toxicity, usually involving intermitotic cells in the skin and mucosa and late complications involving postmitotic cells, including fibrosis, atrophy, necrosis, and vascular damage in tissues with slow turnover, such as the liver, kidney, heart, brain, muscle, and bone. Furthermore, radiotherapy causes secondary malignancies [4]. Additionally, chemotherapy presents a long-term sequelae and side-effects, thus the modern treatments’ goal is to reduce such adverse effects while maintaining efficacy and offering increased survival. These adverse effects include both oral and gastrointestinal mucositis. They may cause local ulceration and pain, which in turn may lead to anorexia, malabsorption, weight loss, anemia, fatigue, and an increased risk of sepsis [5].

The increased knowledge of free radicals and reactive oxygen species (ROS) is producing a new age of disease management. Intriguingly, ROS can also be used in the treatment of diseases because ROS levels can induce tumor cell death [6].

The aim of modern medicine is to find an alternative anticancer treatment in order to avoid the adverse side effects of conventional therapies and, most of all, to find a treatment that is specific for cancer cells without damaging healthy cells.

5-aminolevulinic acid-photodynamic therapy (5-ALA-PDT) is a very selective therapeutic option for a variety of cutaneous and internal malignancies. It spares patients many of the side effects connected with chemotherapy, radiation, and surgery. Additionally, PDT generally does not confer tumor resistance. PDT is regularly used at a few institutions, but more general use has been constrained by the scarcity of randomized clinical trials. PDT merits further research as a therapy paradigm for several oncologic indications due to recent developments in photosensitizing drugs, light sources, and light deliveries.

PDT has the benefit of being administered before or after chemotherapy, radiation, or surgery, without impairing the efficacy of these treatment techniques. Furthermore, sensitivity to PDT is unaffected by the negative effects of chemotherapy or radiation. These results support the use of PDT for improving anti-tumor immunity, at least as an adjuvant therapy, which may be able to control cancer as a secondary illness. The actual difficulty for PDT is to transfer the positive effects shown in the cell-line-based studies and animal models into a therapeutic method that may be used as a human anticancer human treatment [7].

The first-generation photosensitizers (PS) used in PDT were hematoporphyrin and its derivatives, such as Photofrin. Photofrin is a purified mixture of monomers, dimers, and oligomers of hematoporphyrin and its dehydration products [8]. The disadvantages of hematoporphyrin and Photofrin are low toxicity, poor pharmacokinetic profiles, long-lasting light sensitivity, low depth of tissue penetration (generally up to 1 cm), inability to treat disseminated disease, and prolonged skin photosensitivity [9]. Exposure to a single therapeutic dose made patients photosensitive to direct sunlight up to 4–12 weeks later, implying a significant lifestyle change [10]. For such reasons, research on PDT has been focused on the development of alternative new-generation PSs with improved physical, chemical, and therapeutic properties. Second-generation PSs with peak light absorption in the 650–850 nm range were introduced to improve absorption selectivity and greater light penetration at longer wavelengths [11]. The higher lipophilicity gave the drug a low bioavailability, limiting the clinical applications of porphyrins. New delivery systems (antibodies, nanogold preparations, and liposomes) or prodrug approaches have been introduced to overcome this [12,13]. 5-ALA is a prodrug, a small hydrophilic metabolic precursor of endogenous porphyrins with ionizable parts on each end of the carbon skeleton [14]. It possesses a low bioavailability after oral intake but has demonstrated great success in the topical form, eliminating generalized photosensitivity reactions [15]. It possesses poor passive transport across membranes, which can be eliminated when administered in the hydrochloride form; however, this makes injections painful [16].

Phototherapy with the help of light was used in ancient times as it was assumed that light alone was sufficient for the treatment of cancer. Prior to the 20th century, researchers devoted close attention to PSs and how they interact with light exposure to produce therapeutic effects. PDT involves the intraperitoneal, intravenous, and topical administration of photosensitizing chemicals that, when subjected to the proper wavelength of light radiation, produce reactive oxygen species (ROS). A new and recent advancement in cancer therapy has been made due to the use of PDT. It involves systemic or topical introduction of tumor localizing photosensitizing agents. A high degree of selectivity is the best mode adopted by PDT and it is achieved by the combination of visible light and photosensitizing agents, which, when applied in the presence of molecular oxygen, lead to the formation of cytotoxic intermediates that kill tumor cells [17]. The cytotoxicity of PDT collectively involves apoptosis, autophagy, and necrosis.

The pro-photosensitizing endogenous metabolite 5-ALA, a derivative of amino acids, has attracted significant attention because it is a predecessor of PpIX (Figure 1). The hydrophobic PpIX obtained at this stage enters the process of heme synthesis and contextually becomes a PS [18]. Exogenous 5-ALA is rapidly absorbed by cells and converted to PpIX following a similar pathway.

As seen in the Warburg effect, the ferrochelatase enzyme becomes inactive in cancer cells, resulting in an accumulation of PpIX [19]. As result of this PpIX accumulation, the photochemical reaction starts and the concentration of singlet oxygen and superoxide increases in the cell [20]. 

PDT makes use of light, the photosensitizing agent, and oxygen generation. 5-ALA can be activated by an LED light, lamp (halogen, mercury, or xenon) or laser with a wavelength approximately 635 nm (580 to 740 nm), emitting red fluorescence (620 to 710 nm) [21]. The value of light irradiance strongly influences the cellular capacity [22]. Other factors that may influence the in vitro sensitivity of cancer cells are the components of 5-ALA, concentration of 5-ALA, initial cell density, washing conditions, incubation time, timing of irradiation, wavelength of irradiation, fluence, and duration between irradiation and viability assays [23,24]. During this period, the proliferation of cells treated with 5-ALA-PDT can be influenced by the number of viable sigmoid-shaped cells.

PDT treatment has been authorized by the US food and drug authority for the treatment of lung cancer, esophageal cancer, and a variety of malignant cancers, including head and neck cancers, prostate cancer, and brain tumors. Research has shown that in PDT, the photosensitizing agent can be administered repeatedly without causing major complications because it makes use of non-ionizing radiation and does not harm the DNA.

Targeted PDT is another development in PDT treatment. Cancer targeting agents, e.g., monoclonal antibodies, are combined with a PS group to form a phototoxic compound. PSs assemble in the tissue where the cancer marker is over expressed, thus reducing the effect of treatment on nearby neighboring cells. Since the receptor density varies from tumor types to tumor areas, the effectiveness of this therapy is less than PDT treatment.

In certain types of tumors, such as esophageal cancer, levels of surface receptors and signaling molecules increase. Surface receptors are considered the best choice for targeted therapies. Epidermal growth factor receptor is a major receptor that is over expressed in tumor cells. Subsequently, another effective therapy was introduced in which the availability of an increased amount of receptor cells on the surface of cells can lead to the binding of PS compounds to target cancer cells. This can be achieved with the help of tyrosine kinase inhibitors TKIs. TKIs modulate downstream signaling pathways and also help in the over expression of EGFR receptors on the surface of cells [25].

There was a need to introduce a treatment option that is more effective, does less harm to healthy cells, and has low side effects and is less painful [26].

The first successful experiment of PDT was carried out on animals in 1975 at Roswell Park Cancer Institute in Buffalo. From this successful experiment, PDT gained some importance and was used by scientists in experimental models.

Research is underway to search for tumor targeted therapies for cancer. For example, chemotherapy uses cytotoxic agents to kill dividing normal and cancerous cells. These targeted therapies are directed towards abnormal proteins encoded by mutagenic genes. Thus, there has been a technology shift from cytotoxic therapy to the tumor specific actionable mutation and development of molecularly targeted agents, e.g., mutation analysis by sequential oligonucleotide capture, amplification, and PDT, that are the most promising diagnostic tools for cancers [27].

In this study we will review tumor targeted therapy 5-ALA-PDT, which, due to the increased production of ROS, represents a groundbreaking way in a clinical setting to fight cancer. Studies have shown that a higher accumulation of 5-ALA-derived PpIX in proliferating cells may provide a biologic rationale for the clinical use of 5-ALA-PDT. Other than ROS production, the other main mechanisms of 5-ALA-PDT’s anticancer potential is its ability to modulate the cell cycle and apoptosis. This review prioritizes 5-ALA-PDT’s efficacy in different types of cancers. It has been used with a variety of PS, light or laser sources, both alone and in conjunction with other topical agents, giving good results.

## 2. Inorganic Nutrients in the Biosynthesis of Heme

The metabolism of heme takes place in bone marrow (i.e., in erythroblasts and reticulocytes that still contain mitochondria) and liver cells in eight steps [28]. The first step is 5-ALA formation by the mitochondrial enzyme, ALA synthase. In this step, glycine derived from the diet or from serine, and succinyl-CoA, a component of the Krebs cycle, are condensed to ALA in a reaction that requires vitamin B6 [29]. In the second step, ALA molecules come in the cytosol and yield porphobilinogen (PBG) and water molecules, in the presence of zinc metalloenzyme ALA dehydratase. The third step results in polymerization of four pyrrole units. In the fourth step, the porphyrin ring is formed by cyclization of the tetrapyrrole structure.

This is followed by modification of the ring by decarboxylation and oxidations. Then, protoporphyrinogen is formed and is further oxidized to protoporphyrin (Figure 2). Heme synthesis also requires a functional tricarboxylic acid (TCA) cycle and an oxygen supply. In the final step, a ferrous iron is incorporated into protoporphyrin IX to form a molecule called heme. Biosynthesis of heme has the peculiarity, shared by other pathways as well, that the first step and final three steps take place in the mitochondrion. The central steps take place in the cytosol. PpIX is an organic compound that fluoresces with a bright red color when hit by UV light.

## 3. ROS Production by 5-ALA-PDT

ROS are reactive oxygen species that contain unpaired, highly unstable electrons. They include the superoxide anion radical (O_2_^−^), the hydroxyl radical (OH^−^), and nonradical oxidants, such as hydrogen peroxide (H_2_O_2_) and singlet oxygen (^1^O_2_) [30,31]. ROS are involved in a number of physiological functions, such as cell signaling, cell differentiation, sensing of oxygenated cells, adaptive immunity, elimination of unwanted mitotic cells, and mitochondrial physiology [32,33,34]. Mitochondria and NADPH oxidases are responsible for ROS production. ROS are produced by complex I and III of the electron transport chain, which remove an electron from amino acids, glucose, and lipids, transferring it to O_2_ [35]. Complex III releases O_2_^−^ and H_2_O_2_ into both the intermembrane space (∼80%) and the mitochondrial matrix ∼20 [36]. Most of the O_2_ released in the mitochondrial matrix is dismantled into H_2_O_2_ by manganese superoxide dismutase (MnSOD or SOD2). SODs use metal ions, such as copper, iron, manganese, and zinc as cofactors. There is also SOD1(Cu/ZnSOD) in the cytosol and SOD3 (Cu/ZnSOD) in the extracellular matrix [37]. H_2_O_2_ leaves the mitochondrial membrane through specific proteins of the aquaporin family [38].

H_2_O_2_, a major signaling molecule involved in cancer, can damage DNA, proteins, and lipids after undergoing Fenton chemistry with Fe^2+^ to form OH^−^. It can be reduced and converted to H_2_O and O_2_ by peroxiredoxins (PRX), glutathione peroxidase (GPX), and catalase [39]. ROS are also produced by leukocytes in response to B- and T-cell inflammatory mediators through the transmembrane proteins NADPH oxidase (NOX) [40]. NOXs transfer electrons across biological membranes to produce O_2_^−^ and H_2_O_2_, which can diffuse across the membrane. The release of growth factors activates NOX-generated ROS that act as secondary signaling molecules, inactivating peroxiredoxin 1 (PRX1), ensuring the continued accumulation of H_2_O_2_ [41]. NOX-derived ROS stimulate cell survival, genomic instability, metastasis, invasion, and angiogenesis in many common cancers [42]. There is a delicate balance between the amount of ROS and antioxidants; both high levels of ROS (oxidative stress) and excessively low levels of ROS (reductive stress) are deleterious [43].

5-ALA-PDT is a promising anticancer therapy with ROS generation by PpIX in mitochondria. [44] (Figure 3). Cellular uptake of exogenous ALA into the cytosol occurs through peptide transporter 1 (PEPT1), proton-coupled amino acid 1 transporter (PAT1), taurine transporter (TauT), and GABA transporter 2 (GAT2) [45]. The 5-ALA-induced ROS burst resulted in a loss of MMP, ATP production, and mitochondria-dependent apoptosis through upregulation of BAX and downregulation of BCL-2/BcL-xL. In turn, the decrease in MMP and ATP synthesis stimulates ROS production [46]. It can enhance cytochrome c oxidase (COX) activity of the mitochondrial respiratory chain, which can thus disrupt the Warburg effect, generate intracellular ROS, and induce caspase-dependent apoptosis in cancer cells [19,20,47].

ALA increases heme biosynthesis by increasing heme protein. Through iron ion insertion, ALA converts to protoporphyrin IX, generating heme [48]. ALA-PDT could play a tumor suppressing role in precancerous and oral cancer cells through the ROS/MMPs pathway or by stimulating cells to release TGF-β. However, the potential relationship between TGF-β and ROS in human dysplastic oral keratinocytes cells treated with 5-ALA-PDT remains unknown [49]. Excessive ROS production causes cell death; in fact, singlet oxygen from 5-ALA-PDT can induce necroptosis via RIP-3 in glioblastoma cells [50]. The antitumor action of 5-ALA-PDT is also manifested by stimulating antitumor immunity through activation of heat-shock protein and inhibition of heme oxygenase-1 (HO-1) by hyperthermia and by acting as a radiosensitizer, increasing the delayed production of ROS in the cytoplasm after ionizing irradiation (IR) [51,52]. The combination of IR and 5-ALA may help overcome hypoxia-induced RT resistance in prostate cancer. 5-ALA can promote the IR-induced Fenton reaction by forming superoxide and causing ferroptosis in prostate and esophageal cancer cells [53,54].

ROS generation by 5-ALA-PDT in human cholangiocarcinoma cells was dependent on both incubation time and concentration. The level of ROS gradually increased to 0.5 mM after irradiation as PpIX synthesis after 24 h incubation was saturated above 0.5 mM of 5-ALA [55]. The combination of 5-ALA and ferrous sodium citrate (SFC) resulted in the selective accumulation of PS PpIX and reduced cell viability in a concentration-dependent manner. 5-ALA-SFC enhanced ROS generation and reduced cell viability of the human gastric cancer line MKN45 [56].

## 4. 5-ALA-PDT Treatment in Different Type of Cancers

### 4.1. In Vitro Studies

#### 4.1.1. Colorectal Cancer

Colorectal cancer (CRC) has been reported as the second leading cause of tumor deaths worldwide. One of the dangerous facts associated with colon cancer is the recurrence of tumor cells from the residual cells, leading to metastasis and carcinoma. Although various advancements have been made in cancer therapy, there are still two critical factors that are major threats to human life. They include recurrence of colorectal cancer and metastatic spread of cancer. Depending on the stage of cancer, 25% of patients have metastasis while 50% of patients develop metastasis after their follow up treatment. Therefore, there was a strong need to come up with a treatment option other than conventional radiotherapy or chemotherapy treatment options that can effectively treat tumors at the initial stage with its photochemical property.

PDT is one of the best nonconventional treatment options for CRC, playing a major role in the progression of immune response that helps in the defense mechanism against cancer. In CRC, the rectum opening is commonly used to access the colorectal cancer site. Colonoscopy endoscopes are used to administer the PS and specific wavelength to the target area to activate PS drug.

A study on CRC cell lines SW480 and SW460 evaluated the effect of 5-ALA-PDT treatment on interleukin secretion. In CRC, there is an increased production of interleukins IL-6, IL-8, and IL-10, which cause tumor growth, cell proliferation, metastasis, and angiogenesis [57]. Cell lines were incubated with 1000 μM of 5-ALA for 4 h. Cell viability, evaluated by MTT assay, was decreased in cancer cell lines SW480 and SW460 after 5-ALA-PDT treatment, as compared to the control group, decreasing the concentration of interleukins IL-6 and IL-10. 5-ALA-PDT showed no effect on IL-8 secretion.

#### 4.1.2. Glioma

Among different brain cancers, malignant gliomas are very common and account for nearly half of brain tumors. Conventional treatment methods of human glioma consist of surgery, chemotherapy, and radiotherapy, which sometimes have serious consequences. Due to the cancer’s recurrence, surgery is not a viable option and due to the development of resistance and other serious side effects, chemotherapy and radiotherapy are only partially successful. In a study conducted on human glioma cells, intracellular levels of PpIX after 5-ALA treatment of human glioma cells were evaluated. Rat astrocytes and human glioma cells were pretreated with arsenic trioxide (ATO). The results showed that 5-ALA-PDT showed an increased accumulation of PpIX followed by pretreatment with ATO relative to the control groups. Apoptosis activity and viability of the cells in the glioma cells pretreated with ATO compared to control groups were increased and decreased, respectively. It was also observed that the mRNA expression of the CPOX enzyme in Porphyrin synthesis was increased after pretreatment with 0.1 μM ATO. Thus, 5-ALA-PDT pretreatment with ATO can prove to be an effective strategy to improve 5-ALA-induced FGR and PDT in glioma [58].

#### 4.1.3. Breast Cancer

The effect of 5-ALA-PDT along with light irradiation on adenocarcinoma breast cancer cell lines MCF-7 and hepatocellular carcinoma cell lines HepG2 was studied. 5-ALA-PDT and light irradiation alone were not able to induce cytotoxic effects and DNA damage in the cell line, but when used together with a suitably high concentration, PDT was successful. 5-ALA-PDT promoted cell death in a concentration-dependent manner. Furthermore, 5-ALA-PDT created significant amounts of micronuclei in the MCF-7 and HepG2 cell lines, as seen in binucleated cells. Additionally, cancer cell lines treated with 5-ALA-PDT exhibited significantly greater mean percentages of DNA damage and tail moment ratio compared to untreated cells, cells treated with 5-ALA or laser irradiation separately, or cells that had not been treated at all. In comparison to untreated cells or cells treated with 5-ALA or laser irradiation alone, MCF-7 and HepG2 cells treated with 5-ALA for 4 h and then exposed to laser irradiation for 4 min and PDT enhanced the percentages of cell death in a highly significant way. After increasing the concentration of 5-ALA from 0.5 mM to 2 mM for MCF-7 cells, the treatment decreased cell viability from 75% to 46%, compared to 88% for non-treated cells. Additionally, by increasing the concentration of 5-ALA in HepG2 cells, the percentage of cell viability was substantially decreased from 86% in untreated cells to 76% at 0.5 mM 5-ALA-PDT and 63% at 2 mM 5-ALA-PDT. Micronuclei were induced and significantly increased in both cancer cell lines at concentrations of 5-ALA of 0.5 mM and 1 mM and PDT treatment. In the comet assay analysis, DNA damage in the MCF-7 and HEPG2 cell lines were measured post-5-ALA-PDT and laser irradiation. DNA damage was measured by tail DNA for both cells and it was seen that tail DNA content increased in both MCF-7 and HEPG2 cell lines as compared to untreated cells [59].

Another research studied the effect of 5-ALA-PDT on human breast cancer cell line MCF-7 and its derivative multi-drug resistant MCF-7/ADR cells to study the effectiveness of 5-ALA treatment on human breast cancer cell lines. Both cell lines were incubated with 5-ALA for 3 h and then irradiated with light. Cell viability was checked with the help of an MTT assay. The accumulation of PpIX concentration in both cell lines came out to be completely different. There was an increased concentration of PpIX in cell line MCF-7, while the concentration was slightly decreased in MCF-7/ADR cells. Cytotoxicity of MCF-7 and MCF-7/ADR cells upon 5-ALA treatment was evaluated. Cell lines were incubated with 1mM 5-ALA and irradiated with different doses of light. Cellular viability was measured with the help of the MTT assay. Higher levels of cytotoxicity were observed in MCF-7 cells, while a low cytotoxic ratio was observed in the MCF-7/ADR cell line. Without 5-ALA treatment, less than 5% of the cells died under the light dose while the 5-ALA treatment reduced 50% of the cell viability in MCF-7 cell lines. The study showed no significant cytotoxic or phototoxic effect over the MCF-7 resistant cell line. In conclusion, 5-ALA treatment was effective against the (wild-type) breast cancer cell line, while the resistant cell lines must have some internal mechanism which made MCF-7 cell lines resistant to 5-ALA treatment that are yet to be explored [60].

Another study focused on the effect of 5-ALA-PDT mediated LED laser lights for the source of PDT application. LED irradiation was used on two breast cancer cell lines: MCF-7 and MDA-MB-231 [61]. The effectiveness of this therapy was checked by WST-1, annexin V, and acridine orange stain. Cell lines were first treated with 1 mM 5-ALA for 4 h and then irradiated with different doses of energy (6, 9, 12, 18, 24, and 30 J/cm^2^). Results suggested that the cell viability was considerably decreased with post-5-ALA-PDT treatment using LED irradiation with higher doses. Overall, the cell viability decreased more in MDA-MB-231 than MCF-7 cell lines. These results prove that irradiation of 5-ALA-PDT using an LED system can be considered as a good option soon for the treatment of breast cancer.

#### 4.1.4. Bladder Cancer

The effect of 5-ALA-PDT combined with tumor-necrosis-factor-related apoptosis inducing ligand (TRAIL) was studied on bladder cancer cell lines [62]. The individual and combined effect of these drugs were studied on three bladder cancer cell lines: SW780, 647V, and T24. These cell lines were treated with 5–100 ng/mL TRAIL for 18 h and the cytotoxic effect of TRAIL and 5-ALA-PDT was observed in all these cell lines. SW780 showed a sensitive nature to TRAIL, while 647 V and T24 showed resistance to TRAIL. Cytotoxic concentrations of TRAIL from 5–50 ng/mL were effective against all bladder cancer cell lines, while an increase in concentration had no effect on the cytotoxic nature of the cell lines. In the next step, bladder cancer cell lines were pretreated with low-dose 5-ALA (5–50 μM) for 4 h and with visible light (400–750 nm, 7.5 J/cm^2^). The photocytotoxicity of 5-ALA with visible light resulted in decreased cell proliferation. Then, cell lines were incubated with TRAIL for 18 h. At the end, the combined effect of pretreated 5-ALA-PDT and TRAIL were observed on bladder cancer cell lines. Bladder cancer cell lines were pretreated with low-dose 5-ALA-PDT (25 μM) and visible light (400–750 nm, 7.5 J/cm^2^) for 3 h, and cell lines were then incubated with TRAIL for 18 h. The therapy of 5-ALA-PDT along with TRAIL induced cell death in all bladder cancer cell lines and these results showed the efficacy of a combined therapy of 5-ALA-PDT and TRAIL for the treatment of bladder cancer.

#### 4.1.5. Esophageal Cancer

5-ALA-PDT treatment along with specific tyrosine kinase inhibitors TKIs were investigated on esophageal carcinoma cell lines ECA-109 [63]. In this study, ECA-109 cells were incubated with a medium containing EGFR tyrphostin AG1478 or PI3K inhibitor LY294002 (small synthetic compound molecules with specific inhibition of EGFR activation) and then with 5-ALA, and the cells were irradiated with the laser 6 h later. The cell viability was measured with the MTT assay, and the migration ability was detected 24 h post-5-ALA-PDT.

PDT can stimulate a variety of signaling pathways in tumor cells. Esophageal carcinoma shows a high EGFR expression, so it was thought that an appropriate therapy is the use of EGFR inhibitors. PI3K/AKT is an important downstream signaling pathway. The study found out that TKIs blocked the pi3k/AKT and PKB/ AKT signaling pathways, blocked EGFR over expression, inhibited cell proliferation, and hampered tumor growth. The migration and proliferation ability of cell lines were measured by the MTT assay, which revealed that 5-ALA-PDT treatment reduced the cell proliferation and migration of cancer cells. Intriguingly, as the concentration of 5-ALA-PDT treatment continued to increase, a gradual decrease was seen in the concentration of inhibiting cells. Thus, the combination therapy with 5-ALA-PDT and EGFR inhibitors has a synergistic effect on decreasing the proliferation and migration of ECA-109 cells.

#### 4.1.6. Oral Potentially Malignant Disorders and Oral Squamous Cell Carcinoma

PDT, using the topical administration of 5-ALA, can serve as a promising anti-tumor method in the treatment of oral potentially malignant disorders (OPMDs) and oral squamous cell carcinoma (OSCC). The study of Wang shows the development of a high-adhesion-strength dry polyacrylic acid (PAA)-chitosan (CHI)-ALA interpenetrating network hydrogel (PACA) patch that can be applied topically to a moist surface and can easily work as a drug delivery model for 5-ALA [64]. Hydrogels, due to their biocompatibility and adhesion to biotic surface, are currently used in research. The author studied the effect and compared the photo cytotoxicity of 5-ALA and PACA hydrogel adhesives in three dysplastic oral keratinocyte cells: DOK; OSCC cells CAL-27; and normal oral keratinocyte cells HOK. Studies have confirmed that 5-ALA works best when applied topically. Cells were incubated with 5-ALA and PACA hydrogels at a concentration of (20–100 μg/mL). No effect of cell death was observed in cell lines treated with PACA hydrogels. When the cells were treated with light irradiation, prior to PDT treatment, ALA and PDT inhibited the growth of CAL 27 and DOK. DOK cells showed a reduction in cell viability at a concentration of 60–100 μg/mL. Cell viability in DOK decreased with increasing concentration at 100 μg/mL. CAL 27 cells were more susceptible to PDT treatment than DOK cells. Cell viability was decreased more in CAL 27 and DOK cells than HOK cell lines. Cell viability in HOK cells were decreased to a much lesser extent. The study revealed that PACA-PDT induces apoptosis in DOK cells in vitro. DOK cells were treated with 100 μg/mL PACA hydrogels and illuminated with a 653 nm laser light. Results were checked by flow cytometry. DOK cells were treated with PACA-PDT and then 5-ALA-PDT. Results showed that PACA hydrogels without PDT could not induce apoptosis in cells. PDT irradiation was mandatory for the apoptosis to happen in a cell. The proportion of apoptosis was much higher in the PACA-PDT group than in the 5 ALA-PDT group alone. In conclusion, the authors created a new dry mucoadhesive hydrogel-loaded ALA to provide increased PDT for OPMDs. The results showed that the PACA hydrogel has excellent biocompatibility, wet adhesion qualities, and sustained ALA diffusion, indicating that it may be used as an oral mucosal medication delivery. PACA hydrogel-mediated PDT demonstrated strong anticancer activity in OPMDs under exposure to 635 nm laser irradiation in vitro. Dry hydrogel patches would also offer a convenient substitute for conventional approaches for the clinical treatment of OPMDs. These patches have a high comfort and acceptability [64].

#### 4.1.7. Skin Cancer

The main aim of the study by Chi et al. was to check the effectivity of 5-ALA combined with gold nanoparticles on cutaneous squamous cell carcinoma (CSCC) [65]. It is one of the most dangerous cancers and the only possible and standard treatment for localized CSCC is surgical excision, cryosurgery, and desiccation, but due to numerous side effects of these treatments, they are not preferred. Among the non-surgical options, PDT is preferred because of better cosmetic outcomes, low morbidity, and is considerably less harmful than chemotherapy and radiotherapy. 5-ALA-PDT treatment alone in clinical settings is shown to have good results but when combined with some other agents, its effectiveness increases to a higher level. For this purpose, gold nanoparticles (GNP) were designed and used with 5-ALA-PDT treatment.

GNP has characteristic features making it useful in drug delivery systems. Due to its greater surface area, good biocompatibility, and surface chemistry, GNP is proven to have significant results. Nanoparticles, due to their size, enhanced permeability, and retention effect, can be delivered in tissues and transferred to the target area. The hydrophilic nature of 5-ALA-PDT can restrict its ability to be absorbed at the target site. Thus, GNP along with 5-ALA can be used as carriers for PS agents to be delivered on to the affected area. Chi et al. studied the effect of treatment on A431 and HaCat cell lines with 5-ALA, GNPs, and 5-ALA-GNPs illuminated with LED light for 1.5 h. Treatment with 5-ALA and GNP alone had no effect on the cell viability. The MTT assay also confirmed that the cell proliferation ability of the cell lines was not affected at all. Cell lines were then treated with a conjugate of 5-ALA-GNP. PDT treatment with 5ALA and ALA-GNP increased apoptosis in both cell lines. In A431 cell lines, apoptotic rates of PDT treatment with 5-ALA and 5 ALA-GNP were much higher than the treatment of PDT with 5-ALA used alone. In HaCat cells, PDT with 5-ALA and 5-ALA-GNPs suppressed the cell viability without no visible difference between the two treatments. On A431 cells, 5-ALA-PDT and 5-ALA-GNPs treatment showed inhibitory effects much higher than HaCat cell lines. The effect of 5-ALA-GNP-PDT on mRNA and protein expression was determined by QRT-PCR. In A431 cell lines, PDT treatment with 5-ALA-GNPs significantly decreased the expression of the stat3 and bcl2 pathways and increased protein expression of Bax. The cell invasion and migration of cells were determined by migration assay and trans-well invasion assay. PDT with 5-ALA and 5-ALA-GNPs both inhibited cell invasion and migration in A431 cells. The study concluded that 5-ALA-GNP treatment can serve as a promising anti-tumor treatment in cancer patients. Further clinical studies are required to improve the treatment options in cutaneous squamous cell carcinoma. 

Skin cancer with metastatic melanoma (MM) is deadly. In the study by Naidoo et al., the authors developed a PS multicomponent nanoparticle drug conjugate carrier system that specifically targets MM cells via biomarkers in order to enhance MM PDT [66]. An antibody-metalated phthalocyanine-polyethylene glycol-gold nanoparticle therapeutic combination was successfully synthesized and characterized. In vitro-grown MM were used in experiments with PS active medications that target PDT at 673 nm. Results revealed that this medication combination boosted PDT for MM by significantly improving cytotoxic and late apoptotic cellular death in cells as well as PS subcellular localization. The ability to cross biological barriers, ease of functionalization, and ability to trigger photothermal cell death due to their metalated composition are all reasons why gold nanoparticles (AuNPs) can be employed as drug carriers in PDT applications to improve PS passive uptake in tumor cells. A highly sensitive and specific biomarker for MM drug uptake targeting, melanoma inhibitory activity (MIA), is an antigen that is particularly overexpressed on melanoma cells alone. To increase medication solubility and active MM tumor targeting uptake, it would, therefore, seem to be highly ideal to conjugate an MM tumor-targeted antibody, such as anti-MIA, onto a sulpho pure ZnPcS PS carrying AuNPs surface. ZnPcS4 PS medicine was attached onto the surface of amine functionalized AuNPs that also contained Anti-MIA antibodies linked to their surface to actively optimize PS medication administration and increase its uptake and absorption within MM target cancer cells. The outcomes of this study clearly enhanced PDT treatment for this type of skin cancer. These findings suggest that ZnPcS4 is an effective PS for PDT on MM due to its ability to significantly induce cell death. However, the IC50 of 2.5 M ZnPcS4 PS with laser irradiation applied was selected to determine whether the PS, when administered in a drug carrying conjugate, was capable of targeted and improved PDT, as it reported a significant decrease of 38% in cell viability and 48% in cellular cytotoxicity. The results of this work reveal that, in contrast to ZnPcS4 PS drug administration alone, the conjugation of anti-MIA Ab to ZnPcS4-AuNP-PEG5000-SH-NH2 inside the final PS drug conjugate actively and specifically improved ZnPcS4 PS drug uptake in MM cells. Thus, in comparison to control groups, the final PS drug combination saw dramatically increased PDT-induced cytotoxic cell death in MM cancer cells. These outcomes were also linked to AuNPs’ capacity for PDT-induced photothermal cellular death. In addition, compared to ZnPcS4 PS medication administration alone, it was discovered that the combination with PDT was found to have enhanced efficacy for most MM cells.

### 4.2. Clinical Applications of 5-ALA-PDT-In Vivo Studies

#### 4.2.1. Esophageal Cancer and Barrett’s Esophagus

Esophageal cancer is often diagnosed at a later stage of metastasis and thus people are left with no options for surgery, chemotherapy, or radiotherapy. Thus, a safer option of PDT treatment can be applied to have effective results. PDT was approved by US Food and Drug Authority in 1996 for the treatment of esophageal cancer [67,68]. Due to its high tumor specificity and less complications, it is widely used in research and clinical practices.

PDT using first generation PS Photofrin and Lazerphyrin has been adopted widely, but due to its complex composition, poor tissue penetration (Photofrin), and high incidence of hypersensitivity, it no longer possible to use it. Second generation photosensitizing agents, such as 5-ALA, known for their lower toxicity, faster metabolism, and lower phototoxicity, were then introduced.

A study investigated the efficacy and the antitumor effects of 5-ALA-PDT in mice with esophageal cancer (EC) [69]. 5-ALA treatment was applied on EC cells and fluorescence induced by 5-ALA was examined by fluorescence microscope. Cytotoxic effects induced by 5-ALA-PDT were examined with the help of LED lights (red 635 nm, blue 410 nm, and green) to check the best wavelengths effective for antitumor therapy. KYSE150 cells were introduced into the footpad of nude mice to induce popliteal lymph node (PLN) metastasis. 5-ALA-PDT was performed on the footpad once a week for 4 weeks, tumor weights were measured, and PLNs were removed after treatment. The antitumor effects of 5-ALA-PDT were recorded with blue, green, and red LEDs. Blue light PDT treatment was more effective. The results concluded that metastasized PLN was considerably reduced in the ALA group than in the control group.

Another study quoted the effect of PDT using 5-ALA vs. Photofrin on Barrett’s esophagus (BE)-related neoplasia [70]. Five years of randomized controlled trials were conducted after this PDT treatment. Biopsies were taken from Barrett’s esophagus patients within and after clinical trials at 6 weeks, 4 months, and 12 months. Resolution-enhancing white light combination of narrow-band imaging and endoscopy were used. Results were recorded after the completion of PDT treatment. A statistically significant difference in CR-IM (full reversal of intestinal metaplasia) between the groups was discovered during the first endoscopy following the completion of PDT, with ALA being more successful. However, there was no difference between the groups in terms of complete reversal of dysplasia. Soon there was recurrence of dysplasia after PDT treatment. The Kaplan–Meir graph suggested 68% success for 5-ALA-PDT treatment alone, while it showed a 60% success for Photofrin.

Another trial conducted on the patients with recurred neoplasia were treated with final non-PDT therapy. The Kaplan–Meier graph showed a 90% success rate for patients already treated with 5-ALA-PDT and then treated with non-PDT treatment for 5 years. The success rate for Photofrin was comparatively low at 76%. The success rate of CR-IM (complete reversal of intestinal metaplasia) with 5-ALA-PDT was 78%, while the success rate was 63% for Photofrin that showed CR-IM. Therefore, 5-ALA treatment was very much helpful in reversing the neoplasia compared to Photofrin treatment. This trial was implemented with patients who responded less favorably to 5-ALA and with patients with frequent cases of recurred neoplasia. These patients were first treated with 5-ALA-PDT and then were treated with endoscopic mucosal resection (EMR) and radiofrequency ablation (RFA), which are the preferred therapies for patients that are unable to respond to PDT treatment. Results were more promising with this combined treatment for people with dysplastic BE.

#### 4.2.2. Gastric Cancer

When it comes to assessing the size of the tumor and spotting metastatic lesions in gastric cancer, 5-ALA-photodynamic diagnosis (5-ALA-PDD) is a promising and secure diagnostic technique. PDD is a diagnostic method that involves the emission of light-induced excitation fluorescence to enhance early detection, without tumor destruction, after PS exposure to blue light. PDT may also have benefits in terms of reducing the invasiveness of the operation. To confirm the efficacy of the 5-ALA-mediated fluorescence technique for gastric cancer, additional research is required, including a prospective randomized controlled trial. Exogenously supplied 5-ALA is taken up by cells and utilized to create PpIX that exhibits red fluorescence when activated by irradiation of a particular wavelength, primarily visible blue light between 375 and 475 nm. This feature can be used to precisely detect cancer cells since they collect PpIX. PDD is a relatively novel method based on the tumor-specific accumulation of 5-ALA-induced PpIX [71].

In terms of the usage of 5-ALA-PDD in the treatment of gastric cancer, it might be applied as a tool for assessing surgical resection margins and consequently support pathological diagnosis during surgery. Occasionally, judgements must be taken regarding how to proceed with surgery to treat stomach cancer, such as establishing the extent of the disease in patients with hazy margins. In these situations, 5-ALA-PDD can offer helpful data to assess the margins that would be enough for resecting the tumor. The most common type of distant metastasis and post-surgical recurrence in advanced gastric cancer with serosa-invading tumors is peritoneal metastasis from a primary gastric cancer, which is an incurable condition with a dismal prognosis. Since proper staging of gastric cancer, and the identification of peritoneal spread is a requirement for choosing the most suitable therapy, numerous attempts using novel techniques have been performed. Although staging laparoscopy is widely used in the care of patients with advanced gastric cancer to avoid needless laparotomy, it has limitations in terms of visualizing the cancer nest’s spread.

In patients with gastric cancer, staging laparoscopy with 5-ALA-PDD increases the diagnostic precision for peritoneal metastases and is safe. To compare the detection sensitivity of 5-ALA-PDD with that achieved using conventional white light, Kishi et al. studied the utility of 5-ALA-PDD with staging laparoscopy in patients with advanced gastric cancer that had invaded the serosa [72]. In a mouse model of peritoneal metastases, which involved eight mice with 729 peritoneal nodes, they showed that the tumor detection rate of 72% using 5-ALA-induced fluorescence was much higher than that attained using white light (39%). In addition, in 13 patients undergoing staging laparoscopy, 3 metastatic lesions that were undetectable under white light were found with 5-ALA-induced fluorescence. They also compared the outcomes of the 5-ALA-PDD with those of the peritoneal fluid cytology and molecular diagnostic testing in 52 patients.

#### 4.2.3. Head and Neck Cancer

Since head and neck anatomy typically permits simple laser access, early superficial lesions in the larynx, throat, and oral cavity constitute suitable PDT targets. The depth of the laser light’s penetration varies depending on its wavelength. In a summary of findings of the literature, the review from Biel et al. reported an 89% full response rate in patients with carcinoma in situ or early-stage head and neck tumors treated with PDT and Photofrin as a PS [73]. According to preclinical results of Photofrin for head and neck cancer, two-part fractionated light treatment for tumor management may be advantageous. PDT offers patients a functionally appealing alternative.

A study carried out by Ahn et al. presented the experience treating patients with high-risk premalignant and microinvasive head and neck carcinomas with 5-ALA-PDT [74]. The trial used noninvasive optical technologies to check the physiologic and photosensitizing qualities of the tissue to comprehend the relationship between lesion physiology and PDT effectiveness. However, considerably greater levels of oxygenation and blood volume were linked to higher rates of response to PDT in patients with tongue/floor of mouth lesions as well as those who had intact disease at the time of PDT, even though the study did not identify a correlation between blood volume and 3-month complete response. The authors found a relation between poor lesion oxygenation and local recurrences but found no association between lesion oxygenation and marginal recurrence. This phase one study was the first clinical study confirming an association between tumor hypoxia with higher rates of recurrence after PDT, and it lays the foundations for an individualized PDT based on the oxygenation characteristics of the lesions.

Some researchers carried out a study in which 5-ALA was administered orally as a dose of 60 mg/kg diluted in 50 mL water, 4–6 h prior to light delivery. Active signs were checked before, immediately after 5-ALA administration, every 15 min for the first two hours, and then hourly till the surgery [75]. Utilizing a Ceralas Series GaAlAs diode laser, activating light was applied 4–6 h after 5-ALA administration. Participants in the study were illuminated to red light at total fluences of 50, 100, 150, and 200 J/cm^2^ (629–635 nm). All patients received post-treatment instructions to stay out of the sun for three days. The Cooperative Group Common Toxicity Criteria were used to grade toxicity endpoints (CTCAE v.3.0). Patients were monitored weekly for the first three weeks, then every two weeks for the next three months, and then every year after that. Grade 3 toxicity by 30 days following PDT injection was used to spot dose-limiting toxicity (DLT). If there was a recurrence, patients were taken off protocol; however, if there was additional progression, they continued off protocol [75]. A phase one series suggested that PDT is generally well tolerated in the treatment of head and neck malignancies in the premalignant and early stages. The frequency of side effects was anticipated, having an impact on discomfort and mucositis. The patient population did not achieve DLT. Two individuals were ultimately unable to get treatment with ALA-PDT due to Levulan’s influence on triggering hypotensive episodes and/or transaminitis; this led us to carefully choose patients with limited cardiovascular history before proposing PDT with ALA.

Local recurrence patterns initially looked more important among groups with a lower dose than the groups under increasing PDT doses. Local recurrences happened after PDT with rates of 57%, 33%, 25%, and 25% at 50, 100, 150, and 200 J/cm^2^, respectively. These results were complicated by a shorter follow-up for higher-dosage groups; the median follow-up for the lower light dose group was 55 months, but the median follow-up for the higher light dose group was 30 months, with the last three patients who were under observation being followed for 6 months. It is important to highlight those local recurrences that happened between 1.3 and 19.0 months (median) following PDT treatment.

The study demonstrated that when PDT is used, Levulan is generally well tolerated and enables shorter periods of light limitation. We find indications that the treatment, particularly at higher light doses, is successful in terms of local control when function-altering excision is an option. Due to the high recurrences in the past, larger treatment areas should be focused more. More research is required to evaluate local control, particularly when evaluating the effectiveness of lower versus higher PDT light dosages.

#### 4.2.4. Oral Cancer

Tongue cancer (under the category of oral cancer) is one of the most common malignancies in the oral region. Conventional therapies of surgery and chemotherapy have certain limitations due to new advancements in the therapies which have been introduced. One method is the use of 5-ALA–PDT treatment.

This study by Ogasawara et al. examined PpIX fluorescence in tongue tumor tissue to determine the best way to administer 5-ALA [76]. The effect intraperitoneal, oral, and topical administration of 5-ALA-induced PpIX was observed in mouse-transplanted tongue cancer and in normal mice.

In the intraperitoneal mode, 5-ALA in the powdered form was dissolved in 0.2 mL saline and administered at concentration of 250–500 mg kg^−1^ while in oral administration it was given in the same concentration and given by means of a gastric tube. In the case of topical administration, an oil-to-water emulsion was made in which 20% 5-ALA was used. Two derivatives of 5-ALA esters, methyl ester and pentyl ester, were also used and these were compared to the topical application. 5-ALA was administered at time intervals of 1, 2, 3, 4, and 5 h and mice were killed prior to taking samples of the tongue. Fluorescence PpIX emission spectra (10-μm thick) were obtained using a spectrophotometer on a total of five successive frozen slices. The fluorescence microscopic image of the study revealed that red fluorescent light was distributed evenly in tongue tumor tissue 5 h after 5-ALA was administered. There was a weak PpIX accumulation in normal tongue muscles. The tumor group showed higher concentration of 5-ALA-induced PpIX after i.p. and p.o. administration. The control group had lower concentrations of PpIX than the tumor group, which had twice the concentration of PpIX. Maximum PpIX accumulation intensity was seen after 5 h p.o. administration of 5-ALA. Moreover, minimal PpIX accumulation in the tumor was also seen after topical administration of 20% 5-ALA ester derivatives cream. which was like the result of topical administration of 20% 5-ALA cream. Maximum PpIX accumulation in the tongue tumor tissue was seen at 5 h after oral administration of 5-ALA. This study on 5-ALA-PDT on oral cancer suggested that oral administration was the most effective treatment. On the contrary, the topical administration of 20% 5-ALA cream was associated with the lowest PpIX accumulation.

#### 4.2.5. Breast Cancer

Breast cancer is one of the leading cancers in women. One out of eight women are affected by breast cancer. Although many therapies have been suggested for the treatment of breast cancers, i.e., hormonal therapies, surgical mastectomy, and combination therapies, there is still a need to improve treatment options for breast cancer to have more options available to patients. PDT, because of its antitumor effects, advanced technology, and effectiveness, is the latest technology used in clinical practices. A study conducted by Banerjee studied the effect of PDT in primary breast cancer. The 4T1 murine breast carcinoma and 2H11 murine endothelial cells lines were used as an experimental tumor model [77]. Cytotoxicity, vascular epithelial growth factor expression, and apoptosis level and cell migration were checked. An increased cytotoxicity level was observed in 4t1 cell lines, whereas no significant difference in apoptosis level was observed in 2h11 cell lines. Cell viability was reduced to 60% after 5-ALA-PDT and thalidomide (TMD) treatment as compared to the control group that showed 100% viability. 5-ALA-PDT along with TMD treatment were used on the 2h11 cell line. TMD therapy decreased the VEGF expression in 2h11 cell lines. Migration ability of cells was evaluated through the wound healing assay. The motility inhibition of cells with 5-ALA-PDT with TMD treatment was increased from 20% to 24%. Combination therapy of 5-ALA-PDT with TMD increased the apoptotic activity and decreased VEGF expression, which in turn increased the effectiveness of PDT treatment [77].

#### 4.2.6. Brain Cancer

The effectiveness and safety of PDT administration to both healthy and cancerous brain tissue have been assessed in animal and human studies [78]. Another study utilized an experimental orthotopic rat glioma model and reported the anti-tumor activity of 5-ALA-PDT [62]. In another study, U-105MG glioma and CH-157MN meningioma cells were treated for 24 h with 5-ALA and then exposed to 12.4 mW/cm^2^, 11 J/cm^2^ light. Cell viability was measured after 24–48 h from 5-ALA-PDT treatment. Increased cytotoxicity was evaluated to glioma cells and attributed to the preferential accumulation of PpIX within glioma compared to meningioma cells. Repetitive 5-ALA-PDT treatments spaced out over long periods of time (weekly for up to 3 weeks) were found to significantly suppress high-grade gliomas (HGG) spheroid cell development. BT4C HGG tumors were orthotopically implanted in BD-IX rats to test these findings in vivo. At 4–5 h before PDT, 5-ALA (125 or 250 mg/kg) was given intraperitoneally (IP) 3 days after tumor cell implantation. Through the burr hole, PDT was carried out intravenously at the same depth as tumor cell implantation. Light at 632 nm was provided for 10–30 min at optical output levels of 7.5–30 mW from a 400 m flat cut fiber (4.5–54 J). The rats who had several (i.e., two or three) PDT sessions per week had considerably longer median survival times.

A human HGG spheroid model has also shown promising results for 5-ALA-PDT [79]. In this experiment, 5-ALA was added to human HGG spheroids for 4 h before either 25 or 50 J/cm^2^ of light was administered. More cell death occurred within the spheroid when low fluence rates to the same total fluence were used, which may be due to low fluence rates’ improved ability to store oxygen during PDT and continue the continuing creation of ROS by the photochemical process.

The most recent effectiveness data using a preclinical 5-ALA-PDT rat model were published [80]. Athymic fox1 rnu/rnu male rats were orthotopically injected with human U87MG GBM cells. Five hours later, PDT was given. 5-ALA (100 mg/kg) was given Intraperitoneal 14 days after tumor cell implantation. Under the supervision of an MRI, a 350 m flat cut quartz fiber was implanted into the tumor and used to provide light (635 nm) from a diode laser. The interstitial 5-ALA-PDT at 30 mW-treated animals showed symptoms of increased intracranial pressure (ICP), which was lethal in almost 60% of the animals. In the 4.8 mW group, no fatal or serious side effects were noticed. Although the per-group sample sizes were rather modest, the results revealed that fractionated PDT was superior to a single PDT session because significant tumor necrosis was generated in both the low and high fluence rate groups.

In comparison to spheroids treated with a single PDT session, those treated with numerous PDT sessions exhibited much less development potential. These results serve to emphasize the importance of continuing research on prolonged light delivery for PDT in brain cancer.

Additionally, 5-ALA-PDT’s in vitro cytotoxicity has been contrasted with that of 5-ALA derivatives, such as methyl-ALA (m-ALA), hexyl-ALA (h-ALA), and benzyl-ALA (b-ALA). Various doses of 5-ALA and ALA derivatives (0.025–5.0 mM) were treated with human HGG spheroids for 4 h. After 635 nm light (25 J/cm^2^, 25 mW/cm), 5-ALA and m-ALA (0.05 mM) had comparable cytotoxic effects. However, under the same circumstances, h-ALA and b-ALA produced more cytotoxicity. Further analysis of 5-ALA and h-ALA revealed that, at concentrations 10–20 times lower than those of 5-ALA, PDT with h-ALA produced a cytotoxic response comparable to 5-ALA-induced PDT.

5-ALA-PDT treatment has demonstrated efficacy as a therapeutic option against human glioma. Based on these results, 5-ALA based fluorescence guided resection (FGR) and PDT can prove to be effective treatment options for this kind of cancer. However insufficient PpIX accumulation hinders the application of FGR and PDT onto the target areas of glioma. 5-ALA-induced PpIX has been associated with fluorescence guidance integrated in the biopsy needle as a novel intraoperative marker. This improved glioma surgery but was not conclusive because the presence of extravasated blood cannot be easily circumvented. The excitation light for PpIX with the usual wavelength of 405 nm is readily absorbed by blood [81]. The poor prognosis for patients with malignant glioma prompted later clinical trials to evaluate whether PDT could offer longer survival than previous treatment options. The PS used was mostly Photofrin [82]. 5-ALA-PDT may offer a significant survival advantage [83], largely attributable to the immunogenic nature of PDT-induced cell death. It may also indicate the efficient destruction of stem-like tumor cells by PDT, but that there is also a considerable risk of treatment-related side effects, probably due to the limited tumor selectivity of PS accumulation [84]. Clinical trials are currently in preparation.

The therapeutic affectivity and safety of 5-ALA-PDT therapy has been investigated in clinical trials. PDT can be applied to the resection cavity after surgery or PDT treatment can be done by administering cylindrical diffuser fibers (CDF). These CDFs are types of optical fibers that are safely administered on the target to ensure the effective illumination of the tumor cells. The research of inserting CDFs into the target area is underway while some clinical researchers have practiced this technique so far [85].

#### 4.2.7. Gynecological Neoplasia

Vulval intraepithelial neoplasia (VIN), which is characterized by a dysplasia of the vulva’s squamous epithelium, is a precursor to vulval squamous cell carcinoma. VIN used to be categorized from 1–3 depending on the severity of the dysplasia. Currently, VIN simply stands for high-grade, which encompasses previous grades 2 and 3. One estimate states that between the years 1973 and 2000, the prevalence of VIN increased by 411% [86].

The standard therapeutic approaches of CO_2_ laser ablation and surgical excision, with multifocal illness, are both associated with high rates of cancer recurrence. In addition to scarring, vulval anatomical deformity, and constriction of the vaginal introitus, which can result in psychosexual sickness; surgery can have substantial long-term unfavorable repercussions. Therefore, repeated surgical excision can result in chronic and recurring disease in patients with severe disease. The first non-surgical procedure combined 5-ALA with 630 nm non-laser light. Two out of the first ten women showed a histological reaction after receiving a single treatment with a power density of 50 J cm^−2^. Three out of eight women showed a complete histological response after the single treatment dose was increased to 100 J cm^−2^ to increase the response rate. Pre-treatment analgesia was used to help patients facing some problems in treatment tolerance. Eighteen women got PDT treatment and sixteen of them reported symptom improvement (89%).

In the trial conducted by Dayana, 25 women with a combined total of 111 vulvar intraepithelial neoplasias (VIN) were treated with 10 mL of a topical 20% 5-ALA solution to sensitize the lesions. The patients then had therapy using a total of 57 cycles of laser light with a 635 nm wavelength (100 J cm^−2^). Twenty-seven VIN lesions were present in thirteen women (52%) who also had a complete histological response. Overall, 64% percent of the total 111 lesions had regressed three months after the final PDT treatment. Patients who had one or two lesions mostly received one PDT cycle, while those who had three or more lesions usually underwent two PDT cycles.

Contrary to the abovementioned PDT trials, a combination therapy for VIN consisting of two cycles of topical methylated 5-ALA-PDT followed by an eight-week topical imiquimod treatment attempt was made four weeks apart. The use of a combination therapy was justified by the fact that a brief course of imiquimod would trigger a local immune response, have a direct impact on VIN, and create a more favorable immunological environment for PDT to produce response rates that were higher than those previously reported from imiquimod or PDT trials that used the drug alone. Twenty women with VIN were treated, and a clinical response (defined as at least a 50% reduction in the size of the tumor) of 60% (20% complete and 40% partial) was seen at 26 weeks when the lesion was evaluated. Some women who had only a partial clinical response could undergo completion surgery since the lesions were smaller and easier to remove through surgery.

The potential for damaging the surrounding healthy tissue when administering PDT treatment to the vulva was one of the practical difficulties. Photo sensitizers have been used in studies for VIN in a variety of forms, including cream, gel, solution, intravenous route, and so on, for variable times, with an average of three hours for absorption. On an irregular anatomical surface, such as the vulva, with multifocal lesions or a unifocal lesion in a tricky area within skin folds, it is unlikely to maintain a steady absorption of the PS without leakage into the surrounding normal tissue, increasing the risk of damage to the healthy surrounding skin after the application of light. Low doses of PS are believed to have little effect when light is not present, making them non-toxic to the nearby healthy tissue.

#### 4.2.8. Bladder Cancer

Bladder cancer is considered as the fourth most common cancer in men and the eighth most common cancer in women. Most of the bladder cancer cases are superficial (non-muscle invasive). PDT was first reported in 1975, for the treatment of superficial transitional cell carcinoma of the bladder [87]. 5-ALA-PDT is considered the best treatment for bladder cancer because PS can be administered directly intravesically, which directly targets tumors cells and is not able to spread systemically.

A study presented the findings of a multicenter prospective experiment for the effectiveness of bladder cancer treatment combining transurethral resection (TUR) and PDT with Alasens [88]. Forty-five Russian participants in the trial had a confirmed diagnosis of non-muscle invasive bladder cancer. As an anti-relapse procedure, patients had TUR of the bladder while receiving PDT concurrently and they received Alasens as an intravenous instillation of a 3% solution with a 1.5–2-h exposure (prior to TUR). PDT was performed on a single occasion, immediately following TUR, using a combination of local irradiation on the tumor bed and diffuse irradiation on the entire urinary bladder mucosa. There were no complications and there was good treatment tolerance.

The laser system was employed as the light source of a wavelength of 630 nm for photodynamic therapy. Using a laser delivery fiber optic tool, the entire mucosa of the urinary bladder was exposed to diffuse radiation.

According to the literature, the outcomes for patients with bladder tumors were at least comparable to those for actual adjuvant therapy. We were able to get promising results for ALA-PDT as a result, but larger, longer-term trials are required to confirm them. Intraoperative PDT after TUR reduced the recurrence rate for superficial bladder cancer to 22% at the 1-year follow-up versus 40–80% (according to literature data) for TUR as monotherapy. The results of the study gave promising results for 5-ALA-PDT, suggesting this treatment for people with bladder cancer that is not muscle-invasive, but larger, longer-term trials are required to confirm these results. 

#### 4.2.9. Prostate Cancer

A study carried out in a mouse model investigated the treatment effect of a novel form of radio dynamic therapy (RDT) consisting of radiation combined with 5-ALA and carbamide peroxide [89]. The study showed the promising effects of PDT on both primary and post-radiotherapy prostate cancer and consisted of various combinations of high-energy (15-MV) irradiation radiotherapy (RT), 5-ALA, and carbamide peroxide.

According to the authors, this is the first study to investigate the consequences of this specific set of therapies utilized in RDT, as well as the first to use a small animal model to research radio dynamic treatment using 15 MV radiation.

Human prostate cancer cell line PC-3 was bilaterally injected into the mice’s flanks under the skin. The tumors were permitted 1–2 weeks to at least reach the point at which they could be seen on an MR scan (average volume = 180 mm^3^). The mice were housed at room temperature with a 12 h day/night schedule, unrestricted access to food and water, and were put to death within two weeks of the end of the treatment.

As observed, radiotherapy (RT) resulted in a decrease in tumor development compared to the control group. When compared to RT, RDT significantly slowed the growth of the tumor (24 ± 9% and 21 ± 8%) one and two weeks after treatment, respectively. In contrast, neither the carbamide peroxide + RT group nor the 5-ALA + RT group demonstrated a statistically significant reduction in tumor growth when compared to the RT group. Furthermore, there was no statistically significant difference in tumor growth between the carbamide peroxide alone, 5-ALA alone, or 5-ALA + carbamide peroxide groups compared to the control group, suggesting that peroxide and 5-ALA only significantly slow tumor growth when given in combination with RT.

In addition to producing an effect greater than the sum of its parts, RDT was demonstrated to slow tumor growth when compared to RT alone. Furthermore, for each of the tumor size groups into which the data were sorted, RDT was demonstrated to have an impact on tumor growth delay. The findings, therefore, support further research into how radiation energy affects treatment outcomes and show the potential for using clinically relevant, high-radiation energies in radio dynamic therapy. Further studies are required to identify the mechanism by which the peroxide acts in concert with 5-ALA for the treatment and the optimal dose/fractionation scheme for future clinical applications. The effect produced by the RDT designed in this study also highlights the benefit of combining an oxidizing agent to the photo sensitizer and radiation in this treatment.

#### 4.2.10. Actinic Keratoses/Skin Cancer

Squamous cell carcinoma (SCC) in situ and possibly invasive SCC is thought to be caused by actinic keratoses (AK), especially in people with pale skin tones. Many histological characteristics, genetic tumor markers, and p53 alterations are shared by invasive SCCs and AKs. With these details, controlling AK is crucial to get top-notch dermatological care.

The study by Tschen et al. aimed to determine the long-term recurrence rate of AK that has improved after five 5-ALA-PDT treatments; to describe the histology of AK lesions that have been treated but do not fully react to 5-ALA-PDT or return in long-term follow-up; and to evaluate the side effects or response of the patients after 5-ALA-PDT treatment (erythema, oedema, stinging/burning, and hypo- and hyper-pigmentation) [90]. The target lesions were further treated twice with a 20% topical 5-ALA solution utilizing a unique Kera-stick applicator. Prior to visible blue light treatment, which was planned 14–18 h following 5-ALA, patients were recommended to avoid immersing the target lesions in water, exposure to bright light, and using sunscreen, cosmetics, moisturizers/emollients, etc. The 5-ALA-treated lesions were then subjected to 10 J cm^2^ of visible blue light (417 ± 4 nm peak) delivered at 10 mW cm^2^ after being softly rinsed with water and made dry. This phase IV long-term follow-up multicenter study involving 110 patients with non-hyperkeratotic AKs showed the safety and efficacy of 5-ALA-PDT as a topical treatment for AKs on the face and scalp. These results were in accordance with two phase III multicenter trials using 5-ALA-PDT in which patients obtained more than 75% clearing of AKs at weeks 8 and 12. With about one-fifth of the AKs returning during the 12-month research period, the current study also evaluated the value of 5-ALA-PDT treatment for AKs in providing long-term clearance of AK lesions [91].

Studies on ultraviolet (UV) carcinogenesis in animal models reveal that systemic or large-area topical 5-ALA-PDT treatment can block the growth of SCCs [92]. Similarly, the results of two recently released small clinical trials suggest that large-area (full face) application of 5-ALA-PDT for as little as 1 h in patients with AKs, followed by photo activation with blue light in a manner identical to that used in this study, results in the resolution of AKs with an efficacy like that demonstrated in this and in the larger registration trials of this therapy. The authors also showed a significant improvement in swallowing, pigmentation, skin quality, and Griffiths score, among other photo-damage-related metrics, in perilesional skin. The outcomes of these studies, along with the observation that ALA-PDT can significantly slow UV carcinogenesis (in animal models), imply that non-metastasizing squamous cell carcinoma (NMSC) incidence is unaltered by 5-ALA-PDT, and that the therapy should be investigated for its potential to slow the rate of development of new AKs and perhaps NMSCs. It is obvious that more research is required to address these issues.

## 5. Conclusions

5-ALA is the early endogenous precursor in the heme biosynthetic pathway found in nearly all mammalian cells. It is rapidly converted to PpIX, which accumulates in the cancer cells deficient in the ferrochelatase enzyme. As a consequence, a photochemical reaction starts, and the concentration of singlet oxygen and superoxide increases in the cell. The 5-ALA-induced ROS burst resulted in the loss of MMP, ATP production, and mitochondria-dependent apoptosis through upregulation of BAX and downregulation of BCL-2. Excessive ROS production causes cell death. For its ability to induce the production of endogenous porphyrins, 5-ALA can be used for many diagnostic and therapeutic uses, making it a promising PDT agent. 

However, the molecular mechanism underlying enhanced 5-ALA-PpIX in tumor cells remains elusive. Over the past two decades, many studies have been conducted to investigate why some tumors exhibit elevated 5-ALA-mediated PpIX and how to increase PpIX levels to better treat the tumor [93]. The increase of 5-ALA-PpIX could directly cause oncogene activation in tumor cells.

This review has reported all currently available data from in vitro studies of 5-ALA-PDT and its clinical applications in cancer treatment (Figure 4). This knowledge has provided the foundation for the clinical application of 5-ALA-PDT for detecting and targeting tumors, particularly in the skin, brain, and bladder.

The first clinical applications date back to 1975 when PDT was first reported for the treatment of superficial transitional cell carcinoma of the bladder. Fortunately, most bladder cancers are superficial and non-muscle invasive, so 5-ALA-PDT is considered the best treatment for this kind of cancer because the PS can be administered directly intravesically, avoiding tumor-cell spread. Some years after, in 1996, PDT was approved by US Food and Drug Authority for the treatment of esophageal cancer, but the best results were obtained using a combined treatment of 5-ALA-PDT with endoscopic mucosal resection (EMR) and radiofrequency ablation (RFA), reaching a high success rate. Head and neck cancer, as well as cutaneous malignancies, represent suitable PDT targets because superficial lesions in the larynx, throat, and oral cavity anatomically permit an easy penetration of the laser light with a up to an 89% full response rate. In tongue cancer, one of the most common malignancies in the oral region, 5-ALA-PDT was applied in mouse-transplanted tongue cancer with no complications and good tolerance.

Repeated treatments of 5-ALA-PDT over time were found to significantly suppress the spheroid cell development of HGG compared to a single PDT treatment. In gynecological neoplasia clinical trials using combined 5-ALA with 630 nm non-laser light, good results in vulval intraepithelial neoplasia (VIN) in terms of response were achieved, with 64% overall regression three months after the final PDT treatment. Among tumors in which non-surgical treatments are preferable, good results have been obtained in CSCC using 5-ALA-PDT combined with gold nanoparticles. It is preferred because of the better cosmetic outcomes and low morbidity and is considerably less harmful than chemotherapy and radiotherapy. Additionally, for skin cancer with MM, the use of GNP employed as drug carriers in PDT enhances the photothermal cell death of cancer cells. Promising effects of PDT on both primary and post-radiotherapy prostate cancer have been observed using a combinations of high-energy irradiation radiotherapy, 5-ALA, and carbamide peroxide.

However, clinical trials are currently underway and longer-term ones are needed to confirm the results obtained so far. PDT treatment has been shown to be a promising treatment for tumor targeting better than conventional chemotherapy and radiotherapy because its cytotoxicity involves apoptosis, autophagy, and necrosis of the selected cancer cells. 5-ALA-PDT received world-wide approval in 1990 for the treatment of dermatological diseases [94]. Then, it was applied for lesions in the genital area and in the oral cavity. In all the cases, its topical administration gives excellent cosmetic results. Moreover, the systemic administration of 5-ALA does not seem to be severely toxic. 5-ALA-PDT has these advantages over conventional treatments: it is noninvasive, well tolerated by patients, it can be applied to patients who refuse surgery or have pacemakers and a bleeding tendency, it can be used as a palliative treatment, and it can be applied repeatedly without cumulative toxicity [95].

The actual difficulty for PDT is to translate the advancements in understanding the effects in the cell-line-based and animal models into a therapeutic method that may be used as an anticancer treatment. The limitations on the use of PDT concern sometimes the effective absorption of PS drugs in the malignancies. This limitation is particularly evident for tumors of internal organs and, in fact, for gastric cancer, 5-ALA-PDD is preferred to 5-ALA-PDT and is applied to assess surgical resection margins during surgery. For this reason, research on PDT has been focused on the development of alternative new-generation PSs with improved physical, chemical, and therapeutic properties. Much progress has been made and much more is likely to be made near future in order to produce more and more personalized therapies.

## Figures and Tables

**Figure 1 ijms-24-08964-f001:**
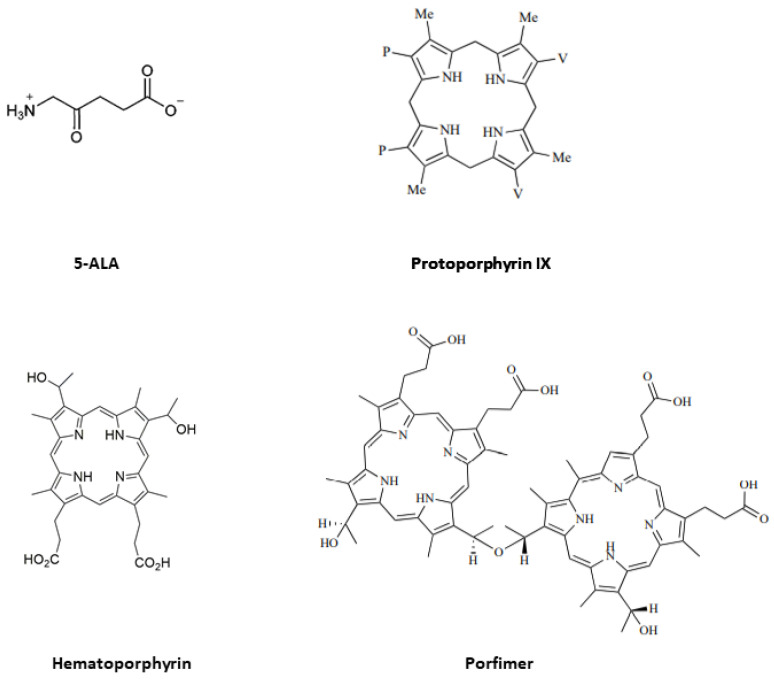
Chemical structures of 5-ALA, protoporhyrin IX, Hematoporphyrin, and Photofrin.

**Figure 2 ijms-24-08964-f002:**
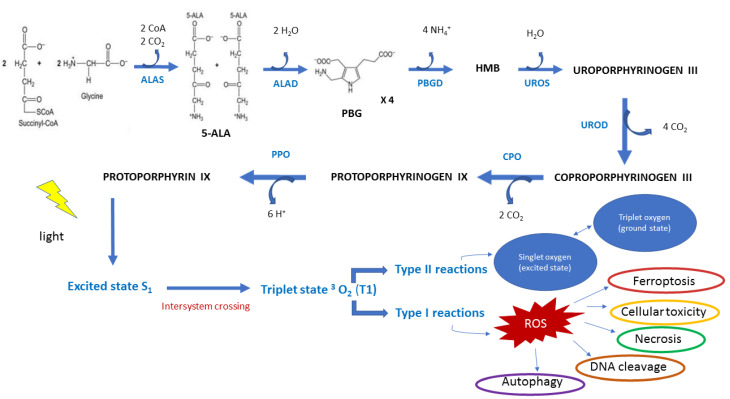
The mechanisms of protoporphyrin IX production and ROS release. In the mitochondrial matrix, succinyl-CoA is condensed with glycine to 5-ALA via the enzyme ALA-synthase (ALAS) and the cofactor pyridoxal-50-phosphate. The 5-amino-4-oxopentanoic acid is converted to the pyrrole derivative porphobilinogen (PBG) through asymmetric condensation attracted by the enzyme ALA-dehydratase (ALAD) in the cytosol. The joining of four PBGs by porphobilinogen deaminase (PBGD) produces 1-hydroxymethylbilane or pre-uroporphyrinogen (HMB), a tetrapyrrolic linear structure. The structure is made cyclic by uroporphyrinogen III cosynthase (UROS). Uporphyrinogen III is transformed into coproporphyrinogen III by uroporphyrinogen decarboxylase (UROD). Coproporphyrinogen III oxidase (CPO) generates protoporphyrinogen IX in the intermembrane space, which undergoes further oxidation in the inner mitochondrial membrane by protoporphyrinogen oxidase (PPO). Activation of PpIX with light of a specific wavelength causes a photodynamic reaction. The PS is excited (S0 to S1). S1 is converted to a more stable triplet state through an intersystem transition. The latter can lose energy by forming highly reactive singlet oxygen species. In the type I reaction, ROS formation occurs, promoting apoptosis, autophagy, ferroptosis, cell toxicity, and necrotic cell death.

**Figure 3 ijms-24-08964-f003:**
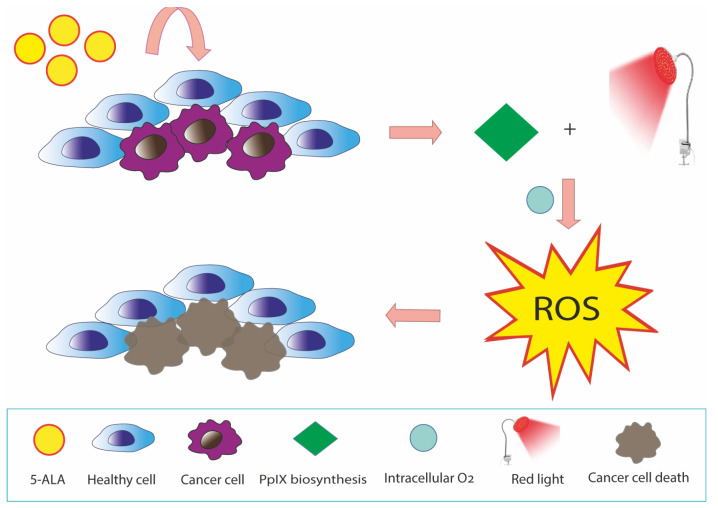
ROS production induced by 5-ALA-PDT in cancer cells. 5-ALA is a pro-photosensitizing endogenous metabolite converted in PpIX in the heme biosynthetic pathway. PpIX accumulation in cancer cells is commonly higher than that in normal cells, as cancer cells are deficient in some enzymes of the pathway. In cancer cells, the presence of molecular oxygen and the excitation of PpIX under irradiation produces ROS that cause DNA and cell membrane injury through lipid peroxidation. This reaction results in damage to mitochondria and other cellular organelles and finally cellular death.

**Figure 4 ijms-24-08964-f004:**
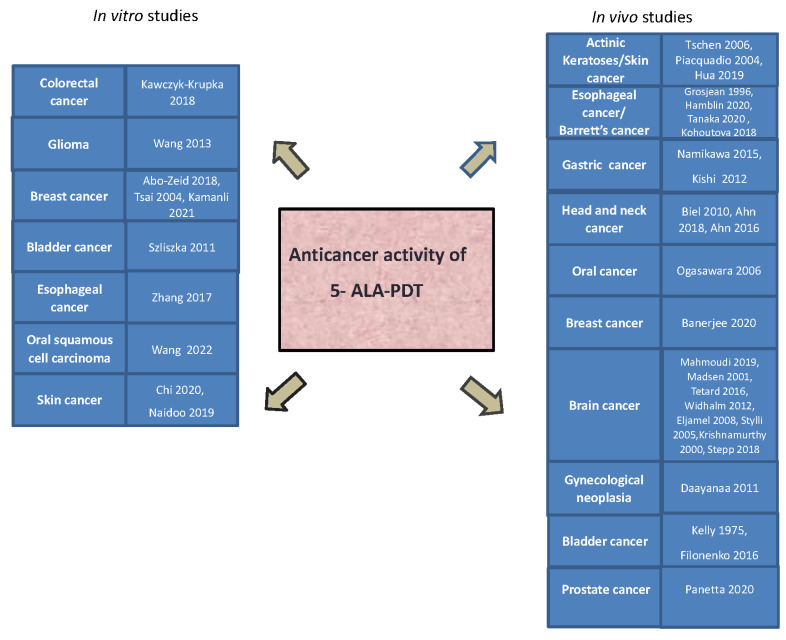
Anticancer effect of 5-ALA-PDT on different cancers. 5-ALA-PDT has been widely investigated for its anticancer activities and low toxicity in several cancers, both in vitro and in vivo [57,58,59,60,61,62,63,64,65,66,67,68,69,70,71,72,73,74,75,76,77,78,79,80,81,82,83,84,85,86,87,88,89,90,91,92].

## Data Availability

Not applicable.

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
