# Peer review of "Reactive Oxygen Species Produced by 5-Aminolevulinic Acid Photodynamic Therapy in the Treatment of Cancer"

_ijms, 2023, doi:10.3390/ijms24108964_

Round 1

Reviewer 1 Report

The introduction needs major revisions:

1) authors need to add chemical structures of 5-ALA, Protoporhyrin IX, Hematoporphyrin, and Photofrin

2) chemically illustrate how 5-ALA get converted into Protoporhyrin IX and how ROS is then generated.

3) authors need to include the limiting factors related to the use of 5-ALA; Hematoporphyrin and Photofrin.

4) authors need to explain how these compounds form prodrugs.

5) provide references for Warburg effect

6) clearly state the light wavelengths (i.e., UVA, UVB, Visible) needed to activate the 5-ALA in PDT.

- Avoid repetitive statements

-use concise language, meaning shorten some of the sentences...they are too long.

Author Response

May 7, 2023

REVIEWER 1 

The introduction needs major revisions:

  • authors need to add chemical structures of 5-ALA, Protoporhyrin IX, Hematoporphyrin, and Photofrin

We thank the reviewer for the suggestions, chemical formulas have been added.

  • chemically illustrate how 5-ALA get converted into Protoporhyrin IX and how ROS is then generated.

We have included a figure explaining the conversion pathway of 5-ALA into Protoporhyrin IX and how photosensitizer from the excited triplet state originates free radicals through Type I and II reactions.

  • authors need to include the limiting factors related to the use of 5-ALA; Hematoporphyrin and Photofrin.

The limitations in the use of first- and second-generation photosensitizers and 5-ALA have been included in the Introduction section.

“Photofrin is a purified mixture of monomers, dimers and oligomers of hematoporphyrin and its dehydration products (Myrzakhmetov B, Arnoux P, Mordon S, Acherar S, Tsoy I, Frochot C. Photophysical Properties of Protoporphyrin IX, Pyropheophorbide-a and Pho-tofrin® in Different Conditions. Pharmaceuticals (Basel). 2021;14(2):138. Published 2021 Feb 9). Disadvantaged of hematoporphyrin and Photofrin are low toxicity , a poor phar-macokinetic profile, long-lasting light sensitivity, low depth of tissue penetration (gener-ally up to 1 cm), inability to treat disseminated disease, prolonged skin photosensitivi-ty(Allison RR, Bagnato VS, Sibata CH. Future of oncologic photodynamic therapy. Future Oncol. 2010;6(6):929-940) . Exposure to a single therapeutic dose made the patient phosensitive to direct sunlight up to 4-12 weeks later, implying a significant lifestyle change (Razum N, Balchum OJ, Profio AE, Carstens F. Skin photosensitivity: duration and intensity following intravenous hematoprohyrin derivatives, HPD and DHE. Photochem Photobiol 1987; 46: 925-8).” And “Second-generation photosensitizers with peak light absorption in the 650-850 nm range were introduced to improve absorption selectivity and greater light penetration at longer wavelengths (Sternberg ED, Dolphin D. Second generation photodynamic agents: a review. J Clin Laser Med Surg. 1993;11(5):233-241). The higher lipophilicity gave the drug low bi-oavailability, limiting the clinical applications of porphyrins. New delivery systems (an-tibodies, nanogold preparations, liposomes) or prodrug approaches have been introduced to overcome this (Casas A, Batlle A. Aminolevulinic acid derivatives and liposome deliv-ery as strategies for improving 5-aminolevulinic acidmediated photodynamic therapy. Curr Med Chem 2006; 13: 1157-68; Oo MK, Yang X, Du H, Wang H. 5-aminolevulinic acid- conjugated gold nanoparticles for photodynamic therapy of cancer. Nanomed 2008; 3: 777-86). 5-ALA is a prodrug, a small hydrophilic metabolic precursor of endogenous por-phyrins with ionizable parts on each end of the carbon skeleton (Oenbrink G, Jurgenlimke P, Gabel D. Accumulation of porphyrins in cells: influence of hydrophobicity aggregation and protein binding, Photochem Photobiol 1988; 48: 451-6). It possesses low bioavailabil-ity after oral intake but has found great success in topical form, eliminating generalized photosensitivity reactions (Lopez RF, Lange N, Guy R, Bentley MV. Photodynamic therapy of skin cancer: controlled drug delivery of 5-ALA and its esters. Adv Drug Deliv Rev. 2004;56(1):77-94). It possesses poor passive transport across membranes, which can be eliminated when administered in hydrochloride form, but that makes injections painful (Musiol R, Serda M, Polanski J. Prodrugs in photodynamic anticancer therapy. Curr Pharm Des. 2011;17(32):3548-3559).”

  • authors need to explain how these compounds form prodrugs.

We have illustrated the endogenous production pathway of 5-ALA (Figure 2).

Because tumors and other proliferating cells tend to exhibit a higher level of PpIX than normal cells after ALA incubation, ALA has been used as a prodrug to enable PpIX fluorescence detection and photodynamic therapy (PDT) of lesion tissues [Yang X, Palasuberniam P, Kraus D, Chen B. Aminolevulinic Acid-Based Tumor Detection and Therapy: Molecular Mechanisms and Strategies for Enhancement. Int J Mol Sci. 2015 Oct 28;16(10):25865-80. doi: 10.3390/ijms161025865. PMID: 26516850; PMCID: PMC4632830].

  • provide references for Warburg effect

The references have been provided.

“Pascale RM, Calvisi DF, Simile MM, Feo CF, Feo F. The Warburg Effect 97 Years after Its Discovery. Cancers (Basel). 2020 Sep 30;12(10):2819; Icard P, Shulman S, Farhat D, Steyaert JM, Alifano M, Lincet H. How the Warburg effect supports aggressiveness and drug resistance of cancer cells? Drug Resist Updat. 2018 May;38:1-11.”

  • clearly state the light wavelengths (i.e., UVA, UVB, Visible) needed to activate the 5-ALA in PDT

The light wavelengths required to activate the 5-ALA have been listed clearly and have been included in the Introduction section.

“The 5-ALA can be activated by an LED light, lamp (halogen, mercury or xenon) or laser of wavelength about 635 nm (580 to 740 nm), emitting red fluorescence (620 to 710 nm) ( Shi-noda, Y.; Kato, D.; Ando, R.; Endo, H. ; Takahashi, T.; Tsuneoka, Y.; Fu-jiwara, Y. Systematic Review and Meta-Analysis of In Vitro Anti-Human Cancer Experiments Investigating the Use of 5-Aminolevulinic Acid (5-ALA) for Photodynamic Thera-py.Pharmaceuticals 2021, 14, 22). The value of light irradiance strongly influences cellular capacity ( Helander, L.; Krokan, H.E.; Johnsson, A.; Gederaas, O.A.; Plaetzer, K. Red versus blue light illumination in hexyl 5-aminolevulinate photodynamic therapy: The influence of light color and irra-diance on treatment outcome in vitro. J. Biomed. Opt. 2014, 19,088002). Other factors that may influence the in vitro sensitivity of cancer cells are the components of 5-ALA, concen-tration of 5-ALA, initial cell density, washing conditions, incubation time, timing of irra-diation, wavelength of irradiation, fluence, and duration between irradiation and viability assays ( Hartl, B. A.; Hirschberg, H.; Marcu, L.; Cherry, S.R. Characterizing low fluence thresholds for in vitro photodynamic therapy. Biomed. Opt. Express 2015, 6, 770-779; Liu, B.; Farrell, T.J.; Patterson, M.S. Comparison of photodynamic therapy with different excita-tion wavelengths using a dynamic model of ami-nolevulinic acid photodynamic therapy of human skin. J. Biomed. Opt. 2012, 17, 088001).”

Reviewer 2 Report

The manuscript in the present form cannot be accepted. It requires many actions.

Chemical structures of the photosensitizers is mandatory.

Tables acollecting the data discussed are mandatory.

Figures totally absent, someoen is mandatory.

All these requests are strictly related to the difficulty for the readership to find data and information useful to fullfill his curiosity. Even if the paper is a review, the topic, very interesting is strictly based on the chemistry of the described events. So hard actions are required before to be considered for eventual publication. 

Authors must decide if photosensitizer is written in this way or as separated words.

Author Response

May 7, 2023

REVIEWER 2 

Comments and Suggestions for Authors

The manuscript in the present form cannot be accepted. It requires many actions.

Chemical structures of the photosensitizers is mandatory.

We provided the requested chemical structures in an additional Figure (Figure 1).

Tables acollecting the data discussed are mandatory.

We provided an additional Figure collecting the data discussed. A figure has been created to provide a better visual impact (Figure 4).

Figures totally absent, someoen is mandatory.

We added three more Figures to the one in the original version. Hopefully these will improve the clarity and the quality of the manuscript.

All these requests are strictly related to the difficulty for the readership to find data and information useful to fullfill his curiosity. Even if the paper is a review, the topic, very interesting is strictly based on the chemistry of the described events. So hard actions are required before to be considered for eventual publication. 

Authors must decide if photosensitizer is written in this way or as separated words.

We corrected the “photosensitizer” word in the entire manuscript.

Totally, we improved manuscript.

Reviewer 3 Report

Numerous anti-cancer therapies make use of reactive oxygen species capacity to kill cancer cells to combat the world's leading cause of death, cancer. Despite improvements in cancer treatment, the currently employed treatment modalities continue to be ineffective, have negative side effects, and require further development in order to become effective. This review focuses on the research that has already been done on 5-ALA-PDT and its effectiveness in treating different cancer pathologies.

Comment 1- It is suggested to begin the manuscript by mentioning the following problem at the start of the manuscript in the introduction section.

Cancer is the leading cause of death worldwide. It is accountable for around 7.6 million deaths worldwide, which is predicted to increase to 13.1 million by 2030. In spite of advances in the treatment of cancer, currently used treatment modules remain ineffective, and their adverse effect and effective cancer therapy still need to be established.

https://doi.org/10.3390/molecules27186051.

Comment 2- Cancer treatment is a major challenge to modern medicine. After a diagnosis of cancer, a multidisciplinary approach to cancer treatment might be required for the patient. However, a specific ethical and psychological approach is required for the treatment of patients with cancer.

Please, cite the following article.

https://doi.org/10.2174/1871520620666200705220307

Comment 3- Provide the structure of 5-aminolevulinic acid.

Comment 4- Free radicals and other reactive oxygen species have been reported to be implicated in the pathology of several human diseases. Therefore, the supplementation of anti-oxidizing agents is needed to prevent the synthesis and encounter of the action of these reactive oxygen species inside the human body.

Cite the following article.

https://doi.org/10.1016/j.genrep.2020.100966.

Comment 4- There must be a table stating the significant role of 5-aminolevulinic acid photodynamic therapy in various types of cancer inhibition.

Comment 5- There should include a paragraph regarding the clinical study linked with 5-aminolevulinic acid photodynamic therapy in various types of cancer inhibition.

Comment 6- Briefly state the advantage of 5-aminolevulinic acid photodynamic therapy over other FDA-approved anti-cancer strategies.

Comment 7- Include the major mechanisms of 5-aminolevulinic acid photodynamic therapy involved in cancer management other than ROS production.

Comment 8- Careful attention needs to be paid to correct all the grammatical errors, manuscript format, and style.

Please recheck section "4.1.4. bladder cancer".

Comment 9- The conclusions are too few. I think they should be expanded.

Comment 10- The authors have not provided the significance of this study.

 Careful attention needs to be paid to correct all grammatical errors and spelling mistakes.

Author Response

May 7, 2023

REVIEWER 3

Numerous anti-cancer therapies make use of reactive oxygen species capacity to kill cancer cells to combat the world's leading cause of death, cancer. Despite improvements in cancer treatment, the currently employed treatment modalities continue to be ineffective, have negative side effects, and require further development in order to become effective. This review focuses on the research that has already been done on 5-ALA-PDT and its effectiveness in treating different cancer pathologies.

Comment 1- It is suggested to begin the manuscript by mentioning the following problem at the start of the manuscript in the introduction section.

Cancer is the leading cause of death worldwide. It is accountable for around 7.6 million deaths worldwide, which is predicted to increase to 13.1 million by 2030. In spite of advances in the treatment of cancer, currently used treatment modules remain ineffective, and their adverse effect and effective cancer therapy still need to be established.

https://doi.org/10.3390/molecules27186051.

Following the reviewer’s suggestion, we improved the beginning of the Introduction section.

Comment 2- Cancer treatment is a major challenge to modern medicine. After a diagnosis of cancer, a multidisciplinary approach to cancer treatment might be required for the patient. However, a specific ethical and psychological approach is required for the treatment of patients with cancer.

Please, cite the following article.

https://doi.org/10.2174/1871520620666200705220307

We improved the Introduction section and cited the suggested article.

Comment 3- Provide the structure of 5-aminolevulinic acid.

We provided the structure of 5-aminolevulinic acid in an additional Figure (Figure 1).

Comment 4- Free radicals and other reactive oxygen species have been reported to be implicated in the pathology of several human diseases. Therefore, the supplementation of anti-oxidizing agents is needed to prevent the synthesis and encounter of the action of these reactive oxygen species inside the human body.

Cite the following article.

https://doi.org/10.1016/j.genrep.2020.100966 

We added the following statement in the Introduction section:

“The increased knowledge of free radicals and reactive oxygen species ROS is producing a new age of disease management [Lobo V, Patil A, Phatak A, Chandra N. Free radicals, antioxidants and functional foods: Impact on human health. Pharmacogn Rev. 2010 Jul;4(8):118-26. doi: 10.4103/0973-7847.70902. PMID: 22228951; PMCID: PMC3249911]. Intriguingly, ROS can also be used in the treatment of desease because ROS levels can induce tumor cell death”.

I apologize for not citing the suggested article, as I provided a different one.

Comment 4- There must be a table stating the significant role of 5-aminolevulinic acid photodynamic therapy in various types of cancer inhibition.

Following the reviewer’s suggestion and doi.org/10.3390/molecules27186051, we provided a new Figure (Figure 4) showing  the anti-cancer activities of 5-ALA- PDT on different cancers both in vitro and in vivo.  A figure has been created to provide a better visual impact.

Comment 5- There should include a paragraph regarding the clinical study linked with 5-aminolevulinic acid photodynamic therapy in various types of cancer inhibition.

A long paragraph regarding the clinical study linked with 5-aminolevulinic acid photodynamic therapy in various types of cancer inhibition is included in the manuscript.

Comment 6- Briefly state the advantage of 5-aminolevulinic acid photodynamic therapy over other FDA-approved anti-cancer strategies.

Cancer treatment modalities divide into conventional and advanced or modern ones.

5-ALA-PDT is among modern anti-cancer strategies.

We added the following statement in the Conclusions section:

“5-ALA-PDT received world-wide approval in 1990 for the treatment of dermatological diseases [ J C Kennedy , R H Pottier, D C Pross. Photodynamic therapy with endogenous protoporphyrin IX: basic principles and present clinical experience. J Photochem Photobiol B. 1990 Jun;6(1-2):143-8. doi: 10.1016/1011-1344(90)85083-9]. Then it was applied for lesions in the genital area and in the oral cavity. Il all the cases its topical administration gives excellent cosmetic results. Also the systemic administration of ALA does not seem to be severely toxic. 5-ALA-PDT has these advantages over conventional treatments: it is noninvasive, well tolerated by patients, it can be applied to patients who refuse surgery or have pacemakers and bleeding tendency; it can be used as a palliative treatment; and it can be applied repeatedly without cumulative toxicity [Peng Q, Warloe T, Berg K, Moan J, Kongshaug M, Giercksky KE, Nesland JM. 5-Aminolevulinic acid-based photodynamic therapy. Clinical research and future challenges. Cancer. 1997 Jun 15;79(12):2282-308. doi: 10.1002/(sici)1097-0142(19970615)79:12<2282::aid-cncr2>3.0.co;2-o. PMID: 9191516]”.

Comment 7- Include the major mechanisms of 5-aminolevulinic acid photodynamic therapy involved in cancer management other than ROS production.

We added the following statement, including the other mechanisms of 5-ALA-PDT involved in cancer, in the Introduction section:

“Other than ROS production, the other main mechanisms of 5-ALA-PDT in the anti-cancer potentiality are its considerable and plausible ability to modulate cell cycle and apoptosis”.

In the manuscript we have argued only about ROS production as it is pertinent to the topic of the special issue.

Comment 8- Careful attention needs to be paid to correct all the grammatical errors, manuscript format, and style.

Please recheck section "4.1.4. bladder cancer".

We corrected the grammatical errors, manuscript format, and style and  rechecked section "4.1.4. bladder cancer".

Comment 9- The conclusions are too few. I think they should be expanded.

We expanded the conclusions.

Comment 10- The authors have not provided the significance of this study.

We emphasized this point in the Introduction section of the manuscript:

“The aims of modern medicine is that of finding alternative anti-cancer treatments, in order to avoid adverse side effects of the conventional therapies to patients, most of all specific for cancer cells without damaging the healthy ones”.

…….5-aminolevulinic acid-photodynamic therapy (5-ALA-PDT) is a very selective therapeutic option for a variety of cutaneous and internal malignancies. It spares patients many of the side effects connected with chemotherapy, radiation, and surgery. Additionally, PDT generally does not confer tumor resistance.

…………Research is underway to search for tumor targeted therapies for cancer. For example, chemotherapy uses cytotoxic agents to kill dividing normal and cancerous cells. These targeted therapies are directed toward to abnormal proteins encoded by mutagenic genes. Thus there has been a technology shift from cytotoxic therapy to the tumor specific actionable mutation and development of molecularly targeted agents e.g. mutation analysis by sequential oligonucleotide capture, amplification and photodynamic therapy are the most promising diagnostic tools for cancers

In this study we will review tumor targeted therapy 5-ALA-PDT which, due to the increased production of ROS, represents a groundbreaking way in clinical setting to fight cancer. This review prioritizes the ALA-PDT treatment's efficacy in different types of cancers. PDT has been used with a variety of PSs and light or laser sources, both alone and in conjunction with other topicals, giving good results.

Comments on the Quality of English Language

 Careful attention needs to be paid to correct all grammatical errors and spelling mistakes.

We corrected the grammatical errors and spelling mistakes.

Round 2

Reviewer 2 Report

The paper resulted to be improved.